# Shuffling Gradient-Based Methods for Nonconvex-Concave Minimax Optimization

**Quoc Tran-Dinh**

Department of Statistics and Operations Research

The University of North Carolina at Chapel Hill

quoctd@email.unc.edu

**Trang H. Tran**

School of OR and Information Engineering

Cornell University, Ithaca, NY

htt27@cornell.edu

**Lam M. Nguyen**

IBM Research, Thomas J. Watson Research Center

Yorktown Heights, NY

LamNguyen.MLTD@ibm.com

## Abstract

This paper aims at developing novel shuffling gradient-based methods for tackling two classes of minimax problems: *nonconvex-linear* and *nonconvex-strongly concave* settings. The first algorithm addresses the nonconvex-linear minimax model and achieves the state-of-the-art oracle complexity typically observed in nonconvex optimization. It also employs a new shuffling estimator for the "hyper-gradient", departing from standard shuffling techniques in optimization. The second method consists of two variants: *semi-shuffling* and *full-shuffling* schemes. These variants tackle the nonconvex-strongly concave minimax setting. We establish their oracle complexity bounds under standard assumptions, which, to our best knowledge, are the best-known for this specific setting. Numerical examples demonstrate the performance of our algorithms and compare them with two other methods. Our results show that the new methods achieve comparable performance with SGD, supporting the potential of incorporating shuffling strategies into minimax algorithms.

## 1 Introduction

Minimax problems arise in various applications across generative machine learning, game theory, robust optimization, online learning, and reinforcement learning (e.g., [1, 2, 3, 5, 12, 13, 17, 19, 21, 25, 35, 40]). These models often involve stochastic settings or large finite-sum objective functions. To tackle these problems, existing methods frequently adapt stochastic gradient descent (SGD) principles to develop algorithms for solving the underlying minimax problems [4, 13]. For instance, in generative adversarial networks (GANs), early algorithms employed stochastic gradient descent-ascent methods where two routines, each using an SGD loop, ran iteratively [13]. However, practical implementations of SGD often incorporate shuffling strategies, as seen in popular deep learning libraries like TensorFlow and PyTorch. This has motivated recent research on developing shuffling techniques specifically for optimization algorithms [4, 5, 8, 16, 26, 32, 38]. Our work builds upon this trend by developing shuffling methods for two specific classes of minimax problems.

**Problem statement.** In this paper, we study the following minimax optimization problem:

$$\min_{w \in \mathbb{R}^p} \max_{u \in \mathbb{R}^q} \left\{ \mathcal{L}(w, u) := f(w) + \mathcal{H}(w, u) - h(u) \equiv f(w) + \frac{1}{n} \sum_{i=1}^{n} \mathcal{H}_i(w, u) - h(u) \right\}, \quad (1)$$

38th Conference on Neural Information Processing Systems (NeurIPS 2024).

where $f : \mathbb{R}^p \to \mathbb{R} \cup \{+\infty\}$ is a proper, closed, and convex function, $\mathcal{H}_i : \mathbb{R}^p \times \mathbb{R}^q \to \mathbb{R}$ are smooth for all $i \in [n] := \{1, 2, \cdots, n\}$, and $h : \mathbb{R}^q \to \mathbb{R} \cup \{+\infty\}$ is also a proper, closed, and convex function. In this paper, we will focus on two classes of problems in (1), overlapped to each other.

(NL) $\mathcal{H}_i$ is nonconvex in $w$ and linear in $u$ as $\mathcal{H}_i(w, u) := \langle F_i(w), Ku \rangle$ for a given function $F_i : \mathbb{R}^p \to \mathbb{R}^m$ and a matrix $K \in \mathbb{R}^{q \times m}$ for all $i \in [n]$ and $(w, u) \in \text{dom}(\mathcal{L})$.

(NC) $\mathcal{H}_i$ is nonconvex in $w$ and $\mathcal{H}_i(w, \cdot) - h(\cdot)$ is strongly concave in $u$ for all $(w, u) \in \text{dom}(\mathcal{L})$.

Although (NC) looks more general than (NL), both cases can be overlapped, but one is not a special case of the other. Under these two settings, our approach will rely on a *bilevel optimization* approach, where the lower-level problem is to solve $\max_u \mathcal{L}(w, u)$, while the upper-level one is $\min_w \mathcal{L}(w, u)$.

**Challenges.** The setting (NL) is a special case of stochastic nonconvex-concave minimax problems because the objective term $\mathcal{H}(w, u) := \langle F(w), Ku \rangle$ is linear in $u$. It is equivalent to the compositional model (CO) described below. However, if $h$ is only merely convex and not strongly convex (e.g., the indicator of a standard simplex), then $\Phi_0$ in (CO) becomes nonsmooth regardless of $F$'s properties. This presents our first challenge. A natural approach to address this issue, as discussed in Section 2, is to smooth $\Phi_0$. The second challenge arises from the composition between the outer function $h^*$ and the finite sum $F(\cdot)$ in (CO). Unlike standard finite-sum optimization, this composition prevents any direct use of existing techniques, requiring a novel approach for algorithmic development and analysis. The third challenge involves unbiased estimators for gradients or "hyper-gradients" in minimax problems. Most existing methods rely on unbiased estimators for objective gradients, with limited work exploring biased estimators. While biased estimators can be used, they require variance reduction properties (see, e.g., [10]). The setting (NC) faces the same second and third challenges as the setting (NL). Additionally, when reformulating it as a minimization problem using a bilevel optimization approach (3), constructing a shuffling estimator for the "hyper-gradient" $\nabla\Phi_0$ becomes unclear. This requires solving the lower-level maximization problem (2). Therefore, it remains an open question whether shuffling gradient-type methods can be extended to this bilevel optimization approach to address (1). In this paper, we address the following research question:

*Can we efficiently develop shuffling gradient methods to solve* (1) *for both* (NL) *and* (NC) *settings?*

Our attempt to tackle this question leads to a novel way of constructing shuffling estimators for the hyper-gradient $\nabla\Phi_0$ or its smoothed counterpart. This allows us to develop two shuffling gradient-based algorithms with rigorous theoretical guarantees on oracle complexity, matching state-of-the-art complexity results in shuffling-type algorithms for nonconvex optimization.

**Related work.** Shuffling optimization algorithms have gained significant attention in optimization and machine communities, demonstrating advantages over standard SGDs, see, e.g., [4, 5, 8, 16, 26, 32, 38]. Nevertheless, applying these techniques to minimax problems like (1) remains challenging, with limited existing literature (e.g., [3, 8, 11]). Das *et al.* in [8] explored a specific case of (1) without nonsmooth terms $f$ and $h$, assuming strong monotonicity and $L$-Lipschitz continuity of the gradient $\nabla\mathcal{H} := [\nabla_w\mathcal{H}, -\nabla_u\mathcal{H}]$ of the joint objective $\mathcal{H}$. Their algorithm simplifies to a shuffling variant of fixed-point iteration or a gradient descent-ascent scheme, not applicable to our settings. Cho and Yun in [3] built upon [8] by relaxing the strong monotonicity to Polyak-Łojasiewicz (PŁ) conditions. This work is perhaps the most closely related one to our algorithm, Algorithm 2, for the (NC) setting. Note that the method in [3] exploits Nash's equilibrium perspective with a simultaneous update, which is different from our alternative update. Moreover, [3] only considers the noncomposite case with $f = 0$ and $h = 0$. Though we only focus on a nonconvex-strongly-concave setting (NC), our results here can be extended to the PŁ condition as in [3]. Very recently, Konstantinos *et al.* in [11] introduced shuffling extragradient methods for variational inequalities, which encompass convex-concave minimax problems as a special case. However, this also falls outside the scope of our work due to the nonconvexity of (1) in $w$. Again, all the existing works in [3, 8, 11] utilize a Nash's equilibrium perspective, while ours leverages a bilevel optimization technique. Besides, in contrast to our sampling-without-replacement approach, stochastic and randomized methods (i.e. using i.i.d. sampling strategies) have been extensively studied for minimax problems, see, e.g., [9, 14, 15, 18, 22, 23, 31, 37, 42]. A comprehensive comparison can be found, e.g., in [3].

**Contribution.** Our main contribution can be summarized as follows.

(a) For setting (NL), we suggest to reformulate (1) into a compositional minimization and exploit a smoothing technique to treat this reformulation. We propose a new way of constructing shuffling estimators for the "hyper-gradient" $\nabla\Phi_\gamma$ (cf. (10)) and establish their properties.

(b) We propose a novel shuffling gradient-based algorithm (*cf.* Algorithm 1) to approximate an $\epsilon$-KKT point of (1) for the setting (NL). Our method requires $\mathcal{O}(n\epsilon^{-3})$ evaluations of $F_i$ and $\nabla F_i$ under the strong convexity of $h$, and $\mathcal{O}(n\epsilon^{-7/2})$ evaluations of $F_i$ and $\nabla F_i$ without the strong convexity of $h$, for a desired accuracy $\epsilon > 0$.

(c) For setting (NC), we develop two variants of the shuffling gradient method: *semi-shuffling* and *full-shuffling* schemes (*cf.* Algorithm 2). The semi-shuffling variant combines both gradient ascent and shuffling gradient methods to construct a new algorithm, which requires $\mathcal{O}(n\epsilon^{-3})$ evaluations of both $\nabla_w \mathcal{H}_i$ and $\nabla_u \mathcal{H}_i$. The full-shuffling scheme allows to perform both shuffling schemes on the maximization and the minimization alternatively, requiring either $\mathcal{O}(n\epsilon^{-3})$ or $\mathcal{O}(n\epsilon^{-4})$ evaluations of $\nabla_u \mathcal{H}_i$ depending on our assumptions, while maintaining $\mathcal{O}(n\epsilon^{-3})$ evaluations of $\nabla_w \mathcal{H}_i$ for a given desired accuracy $\epsilon > 0$.

If a random shuffling strategy is used in our algorithms, then the oracle complexity in all the cases presented above is improved by a factor of $\sqrt{n}$. Our settings (NL) and (NC) of (1) are different from existing works [3, 8, 11], as we work with general nonconvexity in $w$, and linearity or [strong] concavity in $u$, and both $f$ and $h$ are possibly nonsmooth. Our algorithms are not reduced or similar to existing shuffling methods for optimization, but we use shuffling strategies to form estimators for the hyper-gradient $\nabla \Phi_0$ in (5). The oracle complexity in both settings (NL) and (NC) is similar to the ones in nonconvex optimization and in a special case of (1) from [3] (up to a constant factor).

**Paper outline.** The rest of this paper is organized as follows. Section 2 presents our bilevel optimization approach to (1) and recalls necessary preliminary results. Section 3 develops our shuffling algorithm to solve the setting (NL) of (1) and establishes its convergence. Section 4 proposes new shuffling methods to solve the setting (NC) and investigates their convergence. Section 5 presents numerical experiments, while technical proofs and supporting results are deferred to Supp. Docs.

**Notations.** For a function $f$, we use $\mathrm{dom}\,(f)$ to denote its effective domain, and $\nabla f$ for its gradient or Jacobian. If $f$ is convex, then $\nabla f$ denotes a subgradient, $\partial f$ is its subdifferential, and $\mathrm{prox}_f$ is its proximal operator. We use $\mathcal{F}_t$ to denote $\sigma(w_0, w_1, \cdots, w_t)$, a $\sigma$-algebra generated by random vectors $w_0, w_1, \cdots, w_t$, $\mathbb{E}_t[\cdot] = \mathbb{E}[\cdot | \mathcal{F}_t]$ is a conditional expectation, and $\mathbb{E}[\cdot]$ is the full expectation. As usual, $\mathcal{O}(\cdot)$ denotes Big-O notation in the theory of algorithm complexity.

## 2 Bilevel Optimization Approach and Preliminary Results

Our approach relies on a bilevel optimization technique [9] in contrast to Nash's game viewpoint [24], which treats the maximization as a lower level and the minimization as an upper level problem.

### 2.1 Bilevel optimization approach

The minimax model (1) is split into a *lower-level* (*i.e. a follower*) *maximization problem* of the form:

$$
\begin{aligned}
\Phi_0(w) &:= \max_{u \in \mathbb{R}^q} \big\{ \mathcal{H}(w, u) - h(u) \equiv \tfrac{1}{n} \textstyle\sum_{i=1}^n \mathcal{H}_i(w, u) - h(u) \big\}, \\
u_0^*(w) &:= \operatorname*{argmax}_{u \in \mathbb{R}^q} \big\{ \mathcal{H}(w, u) - h(u) \equiv \tfrac{1}{n} \textstyle\sum_{i=1}^n \mathcal{H}_i(w, u) - h(u) \big\}.
\end{aligned}
\tag{2}
$$

For $\Phi_0$ defined by (2), then the *upper-level* (*i.e. the leader*) *minimization problem* can be written as

$$
\Psi_0^\star := \min_{w \in \mathbb{R}^p} \Big\{ \Psi_0(w) := \Phi_0(w) + f(w) \Big\}. \tag{3}
$$

Clearly, this approach is sequential, and only works if $\Phi_0$ is well-defined, i.e. (2) is globally solvable. Hence, the concavity of $\mathcal{H}(w, \cdot) - h(\cdot)$ w.r.t. to $u$ is crucial for this approach as stated below. However, this assumption can be relaxed to a global solvability of (2) combined with a PŁ condition as in [3].

**Assumption 1** (Basic)**.** *Problems* (1) *and* (3) *satisfy the following assumptions for all* $i \in [n]$:

(a) $\Psi_0^\star := \inf_w \Psi_0(w) > -\infty$.
(b) $\mathcal{H}_i$ *is differentiable w.r.t.* $(w, u) \in \mathrm{dom}\,(\mathcal{L})$ *and* $\mathcal{H}_i(w, \cdot)$ *is concave in* $u$ *for any* $w$.
(c) *Both* $f : \mathbb{R}^p \to \mathbb{R} \cup \{+\infty\}$ *and* $h : \mathbb{R}^q \to \mathbb{R} \cup \{+\infty\}$ *are proper, closed, and convex.*

This assumption remains preliminary. To develop our algorithms, we will need more conditions on $\mathcal{H}_i$ and possibly on $f$ and $h$, which will be stated later. In addition, we can work with a sublevel set

$$
\mathcal{L}_{\Psi_0}(w_0) := \{ w \in \mathrm{dom}\,(\Psi_0) : \Psi_0(w) \leq \Psi_0(w_0) \} \tag{4}
$$

of $\Psi_0$ for a given initial point $w_0$ from our methods. If $u_0^*(w)$ is uniquely well-defined for given $w \in \mathcal{L}_{\Psi_0}(w_0)$, then by the well-known Danskin's theorem, $\Phi_0$ is differential at $w$ and its gradient is

$$\nabla\Phi_0(w) = \nabla_w\mathcal{H}(w, u_0^*(w)) = \tfrac{1}{n}\sum_{i=1}^n \nabla_w\mathcal{H}_i(w, u_0^*(w)). \tag{5}$$

We adopt the term "hyper-gradient" from bilevel optimization to name $\nabla\Phi_0$ in this paper.

## 2.2 Technical assumptions and properties of $\Phi_0$ for nonconvex-linear setting (NL)

(a) ***Compositional minimization formulation.*** If $\mathcal{H}_i(w, u) := \langle F_i(w), Ku \rangle$ as in setting (NL), then (1) is equivalently reformulated into the following *nonconvex compositional minimization* problem:

$$\min_{w \in \mathbb{R}^p} \left\{ \Psi_0(w) := f(w) + \Phi_0(w) = f(w) + h^*\left(\tfrac{1}{n}\sum_{i=1}^n K^T F_i(w)\right) \right\}, \tag{CO}$$

where $h^*(v) := \sup_u\{\langle v, u \rangle - h(u)\}$, the Fenchel conjugate of $h$, and $\Phi_0(w) = h^*(K^T F(w))$. If $h$ is not strongly convex, then $h^*$ is convex but possibly nonsmooth.

(b) ***Technical assumptions.*** To develop our algorithms, we also need the following assumptions.

**Assumption 2.** *$h$ is $\mu_h$-strongly convex with $\mu_h \geq 0$, and $\mathrm{dom}(h)$ is bounded by $M_h < +\infty$.*

**Assumption 3** (For $F_i$). *For setting* (NL) *with $\mathcal{H}_i(w, u) := \langle F_i(w), Ku \rangle$ ($i \in [n]$), assume that*

  (a) *$F_i$ is continuously differentiable, and its Jacobian $\nabla F_i$ is $L_{F_i}$-Lipschitz continuous.*
  (b) *$F_i$ is also $M_{F_i}$-Lipschitz continuous or equivalently, its Jacobian $\nabla F_i$ is $M_{F_i}$-bounded.*
  (c) *There exists a positive constant $\sigma_J \in (0, +\infty)$ such that*

$$\tfrac{1}{n}\sum_{i=1}^n \|\nabla F_i(w) - \nabla F(w)\|^2 \leq \sigma_J^2, \quad \forall w \in \mathrm{dom}(F). \tag{6}$$

Assumption 2 allows $\mu_h = 0$ that also covers the non-strong convexity of $h$. Assumption 3 is rather standard to develop gradient-based methods for solving (1). Under Assumption 3, the finite-sum $F$ is also $M_F$-Lipschitz continuous and the Jacobian $\nabla F$ of $F$ is also $L_F$-Lipschitz continuous with

$$M_F := \max\{M_{F_i} : i \in [n]\} \quad \text{and} \quad L_F := \max\{L_{F_i} : i \in [n]\}. \tag{7}$$

Condition (6) can be relaxed to the form $\tfrac{1}{n}\sum_{i=1}^n \|\nabla F_i(w) - \nabla F(w)\|^2 \leq \sigma_J^2 + \Theta_J\|\nabla\Phi_0(w)\|^2$ for some $\Theta_J \geq 0$, where $\nabla\Phi_0$ is a [sub]gradient of $\Phi_0$ or $\Phi_\gamma$ (its smoothed approximation). Moreover, under Assumption 3, if $\mu_h > 0$, then $\nabla h^*$ is $L_{h^*}$-Lipschitz continuous with $L_{h^*} := \tfrac{1}{\mu_h}$. Thus it is possible (see [9]) to prove that $\Phi_0$ is differentiable, and $\nabla\Phi_0$ is also $L_{\Phi_0}$-Lipschitz continuous with $L_{\Phi_0} := M_h\|K\|L_F + \tfrac{M_F^2\|K\|^2}{\mu_h}$ as a consequence of Lemma 4 when $\gamma \downarrow 0^+$ in Supp. Doc. A.

(c) ***Smoothing technique for lower-level maximization problem*** (2). If $h$ is only merely convex (i.e. $\mu_h = 0$), then (2) may not be uniquely solvable, leading to the possible non-differentiability of $\Phi_0$. Let us define the following convex function:

$$\phi_0(v) := \max_{u \in \mathbb{R}^q} \left\{ \langle v, Ku \rangle - h(u) \right\} = h^*(K^T v). \tag{8}$$

Then, $\Phi_0$ in (2) or (CO) can be written as $\Phi_0(w) = \phi_0(F(w)) = \phi_0\left(\tfrac{1}{n}\sum_{i=1}^n F_i(w)\right)$. Our goal is to smooth $\phi_0$ if $h$ is not strongly convex, leading to

$$\begin{cases} \phi_\gamma(v) := \max_u \left\{ \langle v, Ku \rangle - h(u) - \gamma b(u) \right\}, \\ u_\gamma^*(v) := \arg\max_u \left\{ \langle v, Ku \rangle - h(u) - \gamma b(u) \right\}, \end{cases} \tag{9}$$

where $\gamma > 0$ is a given smoothness parameter and $b : \mathbb{R}^q \to \mathbb{R}$ is a proper, closed, and 1-strongly convex function such that $\mathrm{dom}(h) \subseteq \mathrm{dom}(b)$. We also denote $D_b := \sup\{\|\nabla b(u)\| : u \in \mathrm{dom}(h)\}$. In particular, if we choose $b(u) := \tfrac{1}{2}\|u - \bar{u}\|^2$ for a fixed $\bar{u}$, then $u_\gamma^*(v) = \mathrm{prox}_{h/\gamma}(\bar{u} - K^T v)$.

Using $\phi_\gamma$, problem (CO) can be approximated by its smoothed formulation:

$$\min_{w \in \mathbb{R}^p} \left\{ \Psi_\gamma(w) := f(w) + \Phi_\gamma(w) = f(w) + \phi_\gamma(F(w)) \equiv f(w) + \phi_\gamma\left(\tfrac{1}{n}\sum_{i=1}^n F_i(w)\right) \right\}. \tag{10}$$

To develop our method, one key step is to approximate the hyper-gradient of $\Phi_\gamma$ in (10), where

$$\nabla\Phi_\gamma(w) = \nabla F(w)^T \nabla\phi_\gamma(F(w)) = \tfrac{1}{n}\sum_{i=1}^n \nabla F_i(w)^T \nabla\phi_\gamma(F(w)). \tag{11}$$

Then, $\nabla\Phi_\gamma$ is $L_{\Phi_\gamma}$-Lipschitz continuous with $L_{\Phi_\gamma} := M_h\|K\|L_F + \tfrac{M_F^2\|K\|^2}{\mu_h + \gamma}$ (see Lemma 4).

## 2.3 Technical assumptions and properties of $\Phi_0$ for the nonconvex-strongly-concave setting

To develop our shuffling gradient-based algorithms for solving (1) under the nonconvex-strongly-concave setting (NC), we impose the following assumptions.

**Assumption 4** (For $\mathcal{H}_i$). *$\mathcal{H}_i$ for all $i \in [n]$ in (1) satisfies the following conditions:*

(a) *For any given $w$ such that $(w, u) \in \mathrm{dom}\,(\mathcal{H})$, $\mathcal{H}_i(w, \cdot)$ is $\mu_H$-strongly concave w.r.t. $u$.*

(b) *$\nabla \mathcal{H}_i$ is $(L_w, L_u)$-Lipschitz continuous, i.e. for all $(w, u), (\hat{w}, \hat{u}) \in \mathrm{dom}\,(\mathcal{H})$:*

$$\|\nabla \mathcal{H}_i(w, u) - \nabla \mathcal{H}_i(\hat{w}, \hat{u})\|^2 \leq L_w^2 \|w - \hat{w}\|^2 + L_u^2 \|u - \hat{u}\|^2. \tag{12}$$

(c) *There exist two constants $\Theta_w \geq 0$ and $\sigma_w \geq 0$ such that for $(w, u) \in \mathrm{dom}\,(\mathcal{H})$, we have*

$$\tfrac{1}{n} \sum_{i=1}^{n} \|\nabla_w \mathcal{H}_i(w, u) - \nabla_w \mathcal{H}(w, u)\|^2 \leq \Theta_w \|\nabla_w \mathcal{H}(w, u)\|^2 + \sigma_w^2. \tag{13}$$

*There exist two constants $\Theta_u \geq 0$ and $\sigma_u \geq 0$ such that for all $(w, u) \in \mathrm{dom}\,(\mathcal{H})$, we have*

$$\tfrac{1}{n} \sum_{i=1}^{n} \|\nabla_u \mathcal{H}_i(w, u) - \nabla_u \mathcal{H}(w, u)\|^2 \leq \Theta_u \|\nabla_u \mathcal{H}(w, u)\|^2 + \sigma_u^2. \tag{14}$$

Assumption 4(a) makes sure that our lower-level maximization of (1) is well-defined. Assumption 4(b) and (c) are standard in shuffling gradient-type methods as often seen in nonconvex optimization [9].

**Lemma 1** (Smoothness of $\Phi_0$). *Under Assumptions 2 and 4, $u_0^*(\cdot)$ in (2) is $\kappa$-Lipschitz continuous with $\kappa := \frac{L_u}{\mu_H + \mu_h}$. Moreover, $\nabla \Phi_0$ in (5) is $L_{\Phi_0}$-Lipschitz continuous with $L_{\Phi_0} := (1 + \kappa) L_w$.*

## 2.4 Approximate KKT points and approximate stationary points

(a) ***Exact and approximate KKT points and stationary points.*** A pair $(w^\star, u^\star) \in \mathrm{dom}\,(\mathcal{L})$ is called a KKT (Karush-Kuhn-Tucker) point of (1) if

$$0 \in \nabla_w \mathcal{H}(w^\star, u^\star) + \partial f(w^\star) \quad \text{and} \quad 0 \in -\nabla_u \mathcal{H}(w^\star, u^\star) + \partial h(u^\star). \tag{15}$$

Given a tolerance $\epsilon > 0$, **our goal** is to find an $\epsilon$-approximate KKT point $(\widehat{w}, \widehat{u})$ of (1) defined as

$$r_w \in \nabla_w \mathcal{H}(\widehat{w}, \widehat{u}) + \partial f(\widehat{w}), \quad r_u \in -\nabla_u \mathcal{H}(\widehat{w}, \widehat{u}) + \partial h(\widehat{u}), \quad \text{and} \quad \mathbb{E}\big[\|[r_w, r_u]\|^2\big] \leq \epsilon^2. \tag{16}$$

A vector $w^\star \in \mathrm{dom}\,(\Psi_0)$ is said to be a stationary point of (3) if

$$0 \in \nabla \Phi_0(w^\star) + \partial f(w^\star). \tag{17}$$

Since $f$ is possibly nonsmooth, we can define a stationary point of (3) via a gradient mapping as:

$$\mathcal{G}_\eta(w) := \eta^{-1}\big(w - \mathrm{prox}_{\eta f}(w - \eta \nabla \Phi_0(w))\big), \tag{18}$$

where $\eta > 0$ is given. It is well-known that $\mathcal{G}_\eta(w^\star) = 0$ iff $w^\star$ is a stationary point of (3). Again, since we cannot exactly compute $w^\star$, we expect to find an $\epsilon$-stationary point $\widehat{w}_T$ of (3) such that $\mathbb{E}\big[\|\mathcal{G}_\eta(\widehat{w}_T)\|^2\big] \leq \epsilon^2$ for a given tolerance $\epsilon > 0$.

(b) ***Constructing an approximate stationary point and KKT point from algorithms.*** Our algorithms below generate a sequence $\{\widetilde{w}_t\}_{t \geq 0}^{T}$ such that $\frac{1}{T+1} \sum_{t=0}^{T} \mathbb{E}\big[\|\mathcal{G}_\eta(\widetilde{w}_t)\|^2\big] \leq \epsilon^2$. Hence, we construct an $\epsilon$-stationary point $\widehat{w}_T$ using one of the following two options:

$$\widehat{w}_T := \widetilde{w}_{t_*}, \quad \text{where} \begin{cases} t_* := \mathrm{argmin}\{\|\mathcal{G}_\eta(\widetilde{w}_t)\| : 0 \leq t \leq T\}, & \text{(Option 1) or} \\ t_* \text{ is uniformly randomly chosen from } \{0, 1, \cdots, T\} & \text{(Option 2)}. \end{cases} \tag{19}$$

Clearly, we have $\mathbb{E}\big[\|\mathcal{G}_\eta(\widehat{w}_T)\|^2\big] \leq \frac{1}{T+1} \sum_{t=0}^{T} \mathbb{E}\big[\|\mathcal{G}_\eta(\widetilde{w}_t)\|^2\big] \leq \epsilon^2$. We need the following result.

**Lemma 2.** (a) *If $(w^\star, u^\star)$ is a KKT point of (1), then $w^\star$ is a stationary point of (3). Conversely, if $w^\star$ is a stationary point of (3), then $(w^\star, u_0^*(w^\star))$ is a KKT point of (1).*

(b) *If $\widehat{w}_T$ is an $\epsilon$-stationary point of (3) and $\nabla \Phi_0$ is $L_{\Phi_0}$-Lipschitz continuous, then $(\overline{w}_T, \overline{u}_T)$ is an $\hat{\epsilon}$-KKT point of (1), where $\overline{w}_T := \mathrm{prox}_{\eta f}(\widehat{w}_T - \eta \nabla \Phi_0(\widehat{w}_T))$, $\overline{u}_T := u_0^*(\overline{w}_T)$, and $\hat{\epsilon} := (1 + L_{\Phi_0}\eta)\epsilon$.*

(c) *If $\widehat{w}_T$ is an $\epsilon$-stationary point of (10), then $(\overline{w}_T, \overline{u}_T)$ is an $\hat{\epsilon}$-KKT point of (1), where $\overline{w}_T := \mathrm{prox}_{\eta f}(\widehat{w}_T - \eta \nabla \Phi_\gamma(\widehat{w}_T))$, $\overline{u}_T := u_\gamma^*(F(\overline{w}_T))$, and $\hat{\epsilon} := \max\{(1 + L_{\Phi_\gamma}\eta)\epsilon, \gamma D_b\}$.*

Lemma 2 allows us to construct an $\hat{\epsilon}$-approximate KKT point $(\overline{w}_T, \overline{u}_T)$ of (1) from an $\epsilon$-stationary point $\widehat{w}_T$ of either (3) or its smoothed problem (10), where $\hat{\epsilon} = \mathcal{O}(\max\{\epsilon, \gamma\})$.

## 2.5 Technical condition to handle the possible nonsmooth term $f$

To handle the nonsmooth term $f$ of (1) in our algorithms we require one more condition as in [5].

**Assumption 5.** *Let $\Phi_\gamma$ be defined by (10), which reduces to $\Phi_0$ given by (2) as $\gamma \downarrow 0^+$, and $\mathcal{G}_\eta$ be defined by (18). Assume that there exist two constants $\Lambda_0 \geq 1$ and $\Lambda_1 \geq 0$ such that:*

$$\|\nabla\Phi_\gamma(w)\|^2 \leq \Lambda_0\|\mathcal{G}_\eta(w)\|^2 + \Lambda_1, \quad \forall w \in \mathrm{dom}\,(\Phi_0). \tag{20}$$

If $f = 0$, then $\mathcal{G}_\eta(w) \equiv \nabla\Phi_\gamma(w)$, and Assumption 5 automatically holds with $\Lambda_0 = 1$ and $\Lambda_1 = 0$. If $f \neq 0$, then it is crucial to have $\Lambda_0 \geq 1$ in (20). Let us consider two examples to see why?

(i) If $f$ is $M_f$-Lipschitz continuous (e.g., $\ell_1$-norm), then (20) also holds with $\Lambda_0 := 1 + \nu > 1$ and $\Lambda_1 := \frac{1+\nu}{\nu}M_f$ for a given $\nu > 0$.

(ii) If $f = \delta_{\mathcal{W}}$, the indicator of a nonempty, closed, convex, and bounded set $\mathcal{W}$, then Assumption 5 also holds by the same reason as in Example (i) (see Supp. Doc. A).

## 3 Shuffling Gradient Method for Nonconvex-Linear Minimax Problems

We first propose a new construction using shuffling techniques to approximate the true gradient $\nabla\Phi_\gamma$ in (11) for any $\gamma \geq 0$. Next, we propose our algorithm and analyze its convergence.

### 3.1 The shuffling gradient estimators for $\nabla\Phi_\gamma$

**Challenges.** To evaluate $\nabla\Phi_\gamma(w)$ in (11), we need to evaluate both $\nabla F(w)$ and $F(w)$ at each $w$. However, in SGD or shuffling gradient methods, we want to approximate both quantities at each iteration. Note that this gradient can be written in a finite-sum $\frac{1}{n}\sum_{i=1}^{n}\nabla F_i(w)^T\nabla\phi_\gamma(F(w))$ (see (11)), but every summand requires $\nabla\phi_\gamma(F(w))$, which involves the full evaluation of $F$.

**Our estimators.** Let $F_{\pi^{(t)}(i)}(w_{i-1}^{(t)})$ and $\nabla F_{\hat{\pi}^{(t)}(i)}(w_{i-1}^{(t)})$ be the function value and the Jacobian component evaluated at $w_{i-1}^{(t)}$ respectively for $i \in [n]$, where $\pi^{(t)} = (\pi^{(t)}(1), \pi^{(t)}(2), \cdots, \pi^{(t)}(n))$ and $\hat{\pi}^{(t)} = (\hat{\pi}^{(t)}(1), \hat{\pi}^{(t)}(2), \cdots, \hat{\pi}^{(t)}(n))$ are two permutations of $[n] := \{1, 2, \cdots, n\}$. We want to use these quantities to approximate the function value $F(w_0^{(t)})$ and its Jacobian $\nabla F(w_0^{(t)})$ of $F$ at $w_0^{(t)}$, respectively, where $w_0^{(t)}$ the iterate vector at the beginning of each epoch $t$.

For function value $F(w_0^{(t)})$, we suggest the following approximation at each *inner iteration* $i \in [n]$:

$$\textbf{Option 1:} \qquad F_i^{(t)} := \frac{1}{n}\left[\sum_{j=1}^{i}F_{\pi^{(t)}(j)}(w_{j-1}^{(t)}) + \sum_{j=i+1}^{n}F_{\pi^{(t)}(j)}(w_0^{(t)})\right]. \tag{21}$$

Alternative to (21), for all $i \in [n]$, we can simply choose another option:

$$\textbf{Option 2:} \qquad F_i^{(t)} := \frac{1}{n}\sum_{j=1}^{n}F_j(w_0^{(t)}) = \frac{1}{n}\sum_{j=1}^{n}F_{\pi^{(t)}(j)}(w_0^{(t)}). \tag{22}$$

For Jacobian $\nabla F(w_0^{(t)})$, we suggest to use the following standard shuffling estimator for all $i \in [n]$:

$$\nabla F_i^{(t)} := \nabla F_{\hat{\pi}^{(t)}(i)}(w_{i-1}^{(t)}). \tag{23}$$

For $F_i^{(t)}$ from (21) (or (22)) and for $\nabla F_i^{(t)}$ from (23), we form an approximation of $\nabla\Phi_\gamma(w_0^{(t)})$ as

$$\widetilde{\nabla}\Phi_\gamma(w_{i-1}^{(t)}) := (\nabla F_i^{(t)})^T\nabla\phi_\gamma(F_i^{(t)}) \equiv (\nabla F_i^{(t)})^T K u_\gamma^*(F_i^{(t)}). \tag{24}$$

**Discussion.** The estimator $F_i^{(t)}$ for $F$ requires $n - i$ more function evaluations $F_{\pi^{(t)}(j)}(w_0^{(t)})$ at each epoch $t$. The first option (21) for $F$ uses $2n$ function evaluations $F_i$, while the second one in (22) only needs $n$ function evaluations at each epoch $t \geq 0$. However, (21) uses the most updated information up to the *inner iteration* $i$ compared to (22), which is expected to perform better. The Jacobian estimator $\nabla F_i^{(t)}$ is standard and only uses one sample or a mini-batch at each iteration $i$.

### 3.2 The shuffling gradient-type algorithm for nonconvex-linear setting (NL)

We propose Algorithm 1, a shuffling gradient-type method, to approximate a stationary point of (10).

**Discussion.** First, the cost per epoch of Algorithm 1 consists of either $2n$ or $n$ function evaluations $F_i$, and $n$ Jacobian evaluations $\nabla F_i$. Compare to standard shuffling gradient-type methods, e.g., in [8], Algorithm 1 has either $n$ more evaluations of $F_i$ or the same cost. Second, when implementing

---

**Algorithm 1** (Shuffling Proximal Gradient-Based Algorithm for Solving (10))

---
1: **Initialization:** Choose an initial point $\widetilde{w}_0 \in \operatorname{dom}(\Phi_0)$ and a smoothness parameter $\gamma > 0$.
2: **for** $t = 1, 2, \cdots, T$ **do**
3:    Set $w_0^{(t)} := \widetilde{w}_{t-1}$;
4:    Generate two permutations $\pi^{(t)}$ and $\hat{\pi}^{(t)}$ of $[n]$ (identically or randomly and independently)
5:    **for** $i = 1, \cdots, n$ **do**
6:       Evaluate $F_i^{(t)}$ by either (21) or (22) using $\pi^{(t)}$, and $\nabla F_i^{(t)}$ by (23) using $\hat{\pi}^{(t)}$.
7:       Solve (9) to get $u_\gamma^*(F_i^{(t)})$ and form $\widetilde{\nabla}\Phi_\gamma(w_{i-1}^{(t)}) := (\nabla F_i^{(t)})^T K u_\gamma^*(F_i^{(t)})$.
8:       Update $w_i^{(t)} := w_{i-1}^{(t)} - \frac{\eta_t}{n} \widetilde{\nabla}\Phi_\gamma(w_{i-1}^{(t)})$;
9:    **end for**
10:    Compute $\widetilde{w}_t := \operatorname{prox}_{\eta_t f}(w_n^{(t)})$;
11: **end for**

---

Algorithm 1, we do not need to evaluate the full Jacobian $\nabla F_i^{(t)}$, but rather the product of matrix $(\nabla F_i^{(t)})^T$ and vector $\nabla\Phi_\gamma(F_i^{(t)})$ as $\widetilde{\nabla}\Phi_\gamma(w_{i-1}^{(t)}) := (\nabla F_i^{(t)})^T \nabla\Phi_\gamma(F_i^{(t)})$. Evaluating this matrix-vector multiplication is much more efficient than evaluating the full Jacobian $\nabla F_i^{(t)}$ and $\nabla\Phi_\gamma(F_i^{(t)})$ individually. Third, thanks to Assumption 5, the proximal step $\widetilde{w}_t := \operatorname{prox}_{\eta_t f}(w_n^{(t)})$ is only required at the end of each epoch $t$. This significantly reduces the computational cost if $\operatorname{prox}_{\eta_t f}$ is expensive.

### 3.3 Convergence Analysis of Algorithm 1 for Nonconvex-Linear Setting (NL)

Now, we are ready to state the convergence result of Algorithm 1 in a short version: Theorem 1. The full version of this theorem is Theorem 6, which can be found in Supp. Doc. B.

**Theorem 1.** *Suppose that Assumptions 1, 2, 3, and 5 holds for the setting* (NL) *of* (1) *and $\epsilon > 0$ is a sufficiently small tolerance. Let $\{\widetilde{w}_t\}$ be generated by Algorithm 1 after $T = \mathcal{O}(\epsilon^{-3})$ epochs using arbitrarily permutations $\pi^{(t)}$ and $\hat{\pi}^{(t)}$ and a learning rate $\eta_t = \eta := \mathcal{O}(\epsilon)$ (see Theorem 6 in Supp. Doc. B for the exact formulas of $T$ and $\eta$). Then, we have $\frac{1}{T+1}\sum_{t=0}^{T} \|\mathcal{G}_{\eta_t}(\widetilde{w}_t)\|^2 \leq \epsilon^2$.*

*Alternatively, if $\{\widetilde{w}_t\}$ is generated by Algorithm 1 after $T := \mathcal{O}(n^{-1/2}\epsilon^{-3})$ epochs using two random and independent permutations $\pi^{(t)}$ and $\hat{\pi}^{(t)}$ and a learning rate $\eta_t = \eta := \mathcal{O}(n^{1/2}\epsilon)$ (see Theorem 6 in Supp. Doc. B for the exact formulas). Then, we have $\frac{1}{T+1}\sum_{t=0}^{T} \mathbb{E}[\|\mathcal{G}_{\eta_t}(\widetilde{w}_t)\|^2] \leq \epsilon^2$.*

Our first goal is to approximate a stationary point $w^\star$ of (CO) as $\mathbb{E}[\|\mathcal{G}_\eta(\widehat{w})\|^2] \leq \epsilon^2$, while Algorithm 1 only provides an $\epsilon$-stationary of (10). For a proper choice of $\gamma$, it is also an $\epsilon$-stationary point of (3).

**Corollary 1.** *Let $\widehat{w}_T$ defined by* (19) *be generated from $\{\widetilde{w}_t\}$ of Algorithm 1. Under the conditions of Theorem 1 and any permutations $\pi^{(t)}$ and $\hat{\pi}^{(t)}$, the following statements hold.*

(a) *If $h$ is $\mu_h$-strongly convex with $\mu_h > 0$, then we can set $\gamma = 0$, and Algorithm 1 requires $\mathcal{O}(n\epsilon^{-3})$ evaluations of $F_i$ and $\nabla F_i$ to achieve an $\epsilon$-stationary $\widehat{w}_T$ of (3).*

(b) *If $h$ is only convex (i.e. $\mu_h = 0$), then we can set $\gamma := \mathcal{O}(\epsilon)$, and Algorithm 1 needs $\mathcal{O}(n\epsilon^{-7/2})$ evaluations of $F_i$ and $\nabla F_i$ to achieve an $\epsilon$-stationary $\widehat{w}_T$ of (3).*

*If, in addition, $\pi^{(t)}$ and $\hat{\pi}^{(t)}$ are sampled uniformly at random without replacement and independently, and $\Lambda_1 = \mathcal{O}(n^{-1})$, then the numbers of evaluations of $F_i$ and $\nabla F_i$ are reduced by a factor of $\sqrt{n}$.*

## 4 Shuffling Method for Nonconvex-Strongly Concave Minimax Problems

In this section, we develop shuffling gradient-based methods to solve (1) under the nonconvex-strongly concave setting (NC). Since this setting does not cover the nonconvex-linear setting (NL) in Section 3 as a special case, we need to treat it separately using different ideas and proof techniques.

### 4.1 The construction of algorithm

Unlike the linear case with $\mathcal{H}_i(w, u) = \langle F_i(w), Ku \rangle$ in Section 3, we cannot generally compute the solution $u_0^*(\widetilde{w}_{t-1})$ in (2) exactly for a given $\widetilde{w}_{t-1}$. We can only approximate $u_0^*(\widetilde{w}_{t-1})$ by some $\widetilde{u}_t$. This leads to another level of inexactness in an approximate "hyper-gradient" $\widetilde{\nabla}\Phi_0(w_{i-1}^{(t)})$ defined by

$$\widetilde{\nabla}\Phi_0(w_{i-1}^{(t)}) := \nabla_w \mathcal{H}_{\hat{\pi}^{(t)}(i)}(w_{i-1}^{(t)}, \widetilde{u}_t). \tag{25}$$

There are different options to approximate $u_0^*(\widetilde{w}_{t-1})$. We propose two options below, but other choices are possible, including accelerated gradient ascent methods and stochastic algorithms [6, 20].

($a_1$) **Gradient ascent scheme for the lower-level problem.** We apply a standard gradient ascent scheme to update $\widetilde{u}_t$: *Starting from $s = 0$ with $u_0^{(t)} := \widetilde{u}_{t-1}$, at each epoch $s = 1, \cdots, S$, we update*

$$\widehat{u}_s^{(t)} := \operatorname{prox}_{\hat{\eta}_t h}\big(\widehat{u}_{s-1}^{(t)} + \tfrac{\hat{\eta}_t}{n} \sum_{i=1}^{n} \nabla_u \mathcal{H}_i(\widetilde{w}_{t-1}, \widehat{u}_{s-1}^{(t)})\big), \tag{26}$$

*for a given learning rate $\hat{\eta}_t > 0$. Then, we finally output $\widetilde{u}_t := \widehat{u}_S^{(t)}$ to approximate $u_0^*(\widetilde{w}_{t-1})$.*

To make our method more flexible, we allow to perform either only *one iteration* (i.e. $S = 1$) or *multiple iterations* (i.e. $S > 1$) of (26). Each iteration $s$ requires $n$ evaluations of $\nabla_u \mathcal{H}_i$.

($a_2$) **Shuffling gradient ascent scheme for the lower-level problem.** We can also construct $\widetilde{u}_t$ by a *shuffling gradient ascent scheme*. Again, we allow to run either only *one epoch* (i.e. $S = 1$) or *multiple epochs* (i.e. $S > 1$) of the shuffling algorithm to update $\widetilde{u}_t$, leading to the following scheme: *Starting from $s := 1$ with $\widehat{u}_0^{(t)} := \widetilde{u}_{t-1}$, at each epoch $s = 1, 2, \cdots, S$, having $\widehat{u}_{s-1}^{(t)}$, we generate a permutation $\pi^{(s,t)}$ of $[n]$ and run a shuffling gradient ascent scheme as*

$$\begin{cases} u_0^{(s,t)} := \widehat{u}_{s-1}^{(t)}, \\ \textit{For } i = 1, 2, \cdots, n, \textit{ update} \\ \quad u_i^{(s,t)} := u_{i-1}^{(s,t)} + \tfrac{\hat{\eta}_t}{n} \nabla_u \mathcal{H}_{\pi^{(s,t)}(i)}(\widetilde{w}_{t-1}, u_{i-1}^{(s,t)}), \\ \widehat{u}_s^{(t)} := \operatorname{prox}_{\hat{\eta}_t h}(u_n^{(s,t)}). \end{cases} \tag{27}$$

*At the end of the $S$-th epoch, we output $\widetilde{u}_t := \widehat{u}_S^{(t)}$ as an approximation to $u_0^*(\widetilde{w}_{t-1})$.* Here, we use the same learning rate $\hat{\eta}_t$ for all epochs $s \in [S]$. Each epoch $s$ requires $n$ evaluations of $\nabla_u \mathcal{H}_i$.

(b) **Shuffling gradient descent scheme for the upper-level minimization problem.** Having $\widetilde{u}_t$ from either (26) or (27), we run a *shuffling gradient descent epoch* to update $\widetilde{w}_t$ from $\widetilde{w}_{t-1}$ as

$$\begin{cases} w_0^{(t)} := \widetilde{w}_{t-1}, \\ \text{For } i = 1, 2, \cdots, n, \text{ update} \\ \quad w_i^{(t)} := w_{i-1}^{(t)} - \tfrac{\eta_t}{n} \widetilde{\nabla} \Phi_0(w_{i-1}^{(t)}) \equiv w_{i-1}^{(t)} - \tfrac{\eta_t}{n} \nabla_w \mathcal{H}_{\hat{\pi}^{(t)}(i)}(w_{i-1}^{(t)}, \widetilde{u}_t), \\ \widetilde{w}_t := \operatorname{prox}_{\eta_t f}(w_n^{(t)}). \end{cases} \tag{28}$$

These two steps (26) (or (27)) in $u$ and (28) in $w$ are implemented alternatively for $t = 1, \cdots, T$.

(c) **The full algorithm.** Combining both steps (26) (or (27)) and (28), we can present an *alternating shuffling proximal gradient algorithm* for solving (1) as in Algorithm 2.

---

**Algorithm 2** (Alternating Shuffling Proximal Gradient Algorithm for Solving (1) under setting (NC))

1: **Initialization:** Choose an initial point $(\widetilde{w}_0, \widetilde{u}_0) \in \operatorname{dom}(\mathcal{L})$.
2: **for** $t = 1, 2, \cdots, T$ **do**
3:      Compute $\widetilde{u}_t$ using either (26) or (27).
4:      Set $w_0^{(t)} := \widetilde{w}_{t-1}$ and generate a permutation $\hat{\pi}^{(t)}$ of $[n]$.
5:      **for** $i = 1, \cdots, n$ **do**
6:          Evaluate $\widetilde{\nabla} \Phi_0(w_{i-1}^{(t)}) := \nabla_w \mathcal{H}_{\hat{\pi}^{(t)}(i)}(w_{i-1}^{(t)}, \widetilde{u}_t)$.
7:          Update $w_i^{(t)} := w_{i-1}^{(t)} - \tfrac{\eta_t}{n} \widetilde{\nabla} \Phi_0(w_{i-1}^{(t)})$.
8:      **end for**
9:      Compute $\widetilde{w}_t := \operatorname{prox}_{\eta_t f}(w_n^{(t)})$.
10: **end for**

---

**Discussion.** Algorithm 2 has a similar form as Algorithm 1, where $u_0^*(\widetilde{w}_{t-1})$ is approximated by $\widetilde{u}_t$. In Algorithm 1, $u_0^*(\widetilde{w}_{t-1})$ is approximated by $u_\gamma^*(F_i^{(t)})$. Moreover, Algorithm 1 solves the smoothed problem (10) of (3), while Algorithm 2 directly solves (3). Depending on the choice of method to approximate $u_0^*(\widetilde{w}_{t-1})$, we obtain different variants of Algorithm 2. We have proposed two variants:

- **Semi-shuffling variant:** We use (26) for computing $\widetilde{u}_t$ to approximate $u_0^*(\widetilde{w}_{t-1})$.
- **Full-shuffling variant:** We use (27) for computing $\widetilde{u}_t$ to approximate $u_0^*(\widetilde{w}_{t-1})$.

Note that Algorithm 2 works in an alternative manner, where it approximates $u_0^*(\widetilde{w}_{t-1})$ up to a certain accuracy before updating $\widetilde{w}_t$. This alternating update is very natural and has been widely applied to solve minimax optimization as well as bilevel optimization problems, see, e.g., [1, 9, 13].

### 4.2 Convergence analysis

Now, we state the convergence of both variants of Algorithm 2: *semi-shuffling* and *full-shuffling* variants. The full proof of the following theorems can be found in Supp. Doc. C.

(a) ***Convergence of the semi-shuffling variant.*** Our first result is as follows.

**Theorem 2.** *Suppose that Assumptions 1, 2, 4, and 5 hold for (1), and $\mathcal{G}_\eta$ is defined by (18).*

*Let $\{(\widetilde{w}_t, \widetilde{u}_t)\}$ be generated by Algorithm 2 using the **gradient ascent scheme** (26) with $\eta := \mathcal{O}(\epsilon)$ explicitly given in Theorem 8 of Supp. Doc. C, $\hat{\eta} \in (0, \frac{2}{L_u + \mu_h}]$, $S := \mathcal{O}\big(\frac{1}{\hat{\eta}}\big(\mu_h + \frac{4L_u \mu_H}{L_u + \mu_H}\big)^{-1}\big) = \mathcal{O}(1)$, and $T := \mathcal{O}(\epsilon^{-3})$ explicitly given in Theorem 8. Then, we have $\frac{1}{T+1}\sum_{t=0}^{T}\|\mathcal{G}_\eta(\widetilde{w}_t)\|^2 \leq \epsilon^2$.*

*Consequently, Algorithm 2 requires $\mathcal{O}(n\epsilon^{-3})$ evaluations of both $\nabla_w \mathcal{H}_i$ and $\nabla_u \mathcal{H}_i$ to achieve an $\epsilon$-stationary point $\widehat{w}_T$ of (3) computed by (19).*

Note that Theorem 2 holds for both $S > 1$ and $S = 1$ (i.e. we perform only one iteration of (26)).

(b) ***Convergence of the full-shuffling variant – The case $S > 1$ with multiple epochs.*** We state our results for two separated cases: only $\mathcal{H}_i$ is $\mu_H$-strongly convex, and only $h$ is $\mu_h$-strongly convex.

**Theorem 3** (Strong convexity of $\mathcal{H}_i$). *Suppose that Assumptions 1, 2, 4, and 5 hold, and $\mathcal{H}_i$ is $\mu_H$-strongly concave with $\mu_H > 0$ for $i \in [n]$, but $h$ is only merely convex.*

*Let $\{(\widetilde{w}_t, \widetilde{u}_t)\}$ be generated by Algorithm 2 using $S$ epochs of the **shuffling routine** (27) and fixed learning rates $\eta_t = \eta := \mathcal{O}(\epsilon)$ as given in Theorem 8 of Supp. Doc. C for a given $\epsilon > 0$, $\hat{\eta}_t := \hat{\eta} = \mathcal{O}(\epsilon)$, $S := \big\lfloor \frac{\ln(7/2)}{\mu_H \hat{\eta}} \big\rfloor$, and $T := \mathcal{O}(\epsilon^{-3})$. Then, we have $\frac{1}{T+1}\sum_{t=0}^{T}\|\mathcal{G}_\eta(\widetilde{w}_t)\|^2 \leq \epsilon^2$.*

*Consequently, Algorithm 2 requires $\mathcal{O}(n\epsilon^{-3})$ evaluations of $\nabla_w \mathcal{H}_i$ and $\mathcal{O}(n\epsilon^{-4})$ evaluations of $\nabla_u \mathcal{H}_i$ to achieve an $\epsilon$-stationary point $\widehat{w}_T$ of (3) computed by (19).*

**Theorem 4** (Strong convexity of $h$). *Suppose that Assumptions 1, 2, 4, and 5 hold for (1), and $h$ is $\mu_h$-strongly convex with $\mu_h > 0$, but $\mathcal{H}_i$ is only merely concave for all $i \in [n]$. Then, under the same settings as in Theorem 3, but with $S := \big\lfloor \frac{\ln(7/2)}{\mu_h \hat{\eta}} \big\rfloor$, the conclusions of Theorem 3 still hold.*

(c) ***Convergence of the full-shuffling variant – The case $S = 1$ with one epoch.*** Both Theorems 3 and 4 require $\mathcal{O}(n\epsilon^{-4})$ evaluations of $\nabla_u \mathcal{H}_i$. To improve this complexity, we need two additional assumptions but can perform only one epoch of (27), i.e. $S = 1$.

**Assumption 6.** *Let $\hat{\mathcal{G}}_\eta(u) := \eta^{-1}(u - \mathrm{prox}_{\eta h}(u + \eta \nabla_u \mathcal{H}(w, u)))$ be the gradient mapping of $\psi(w, \cdot) := -\mathcal{H}(w, \cdot) + h(\cdot)$. Assume that there exist $\hat{\Lambda}_0 \geq 1$ and $\hat{\Lambda}_1 \geq 0$ such that*

$$\|\nabla_u \mathcal{H}(w, u)\|^2 \leq \hat{\Lambda}_0 \|\hat{\mathcal{G}}_\eta(u)\|^2 + \hat{\Lambda}_1, \quad \forall (w, u) \in \mathrm{dom}\,(\mathcal{L}). \tag{29}$$

Clearly, if $h = 0$, then $\hat{\mathcal{G}}_\eta(u) = -\nabla_u \mathcal{H}(w, u)$ and (20) automatically holds for $\hat{\Lambda}_0 = 1$ and $\hat{\Lambda}_1 = 0$. Assumption 6 is similar to Assumption 5, and it is required to handle the prox operator of $h$ in (27).

**Assumption 7.** *For $f$ in (1), there exists $L_f \geq 0$ such that*

$$f(y) \leq f(x) + \langle f'(x), y - x \rangle + \frac{L_f}{2}\|y - x\|^2, \quad \forall x, y \in \mathrm{dom}\,(f), \ f'(x) \in \partial f(x). \tag{30}$$

Clearly, if $f$ is $L_f$-smooth, then (30) holds. If $f$ is also convex, then (30) implies that $f$ is $L_f$-smooth.

Under these additional assumptions, we have the following result.

**Theorem 5.** *Suppose that Assumptions 1, 2, 4, 5, 6, and 7 hold and $\mathcal{G}_\eta$ is defined by (18).*

*Let $\{(\widetilde{w}_t, \widetilde{u}_t)\}$ be generated by Algorithm 2 using **one epoch** ($S = 1$) of the **shuffling routine** (27), and fixed learning rates $\eta_t = \eta := \mathcal{O}(\epsilon)$ as in Theorem 9 of Supp. Doc. C for a given $\epsilon > 0$, $\hat{\eta}_t := \hat{\eta} = 30\kappa^2 \eta$, and $T := \mathcal{O}(\epsilon^{-3})$, where $\kappa := \frac{L_u}{\mu_H + \mu_h}$. Then, we have $\frac{1}{T+1}\sum_{t=0}^{T}\|\mathcal{G}_\eta(\widetilde{w}_t)\|^2 \leq \epsilon^2$.*

*Consequently, Algorithm 2 requires $\mathcal{O}(n\epsilon^{-3})$ evaluations of both $\nabla_w \mathcal{H}_i$ and of $\nabla_u \mathcal{H}_i$ to achieve an $\epsilon$-stationary point $\widehat{w}_T$ of (3) computed by (19).*

Similar to Algorithm 1, if $\pi^{(s,t)}$ and $\hat{\pi}^{(t)}$ are generated randomly and independently, $\Lambda_1 = \mathcal{O}(1/n)$, and $\hat{\Lambda}_1 = \mathcal{O}(1/n)$, then our complexity stated above can be improved by a factor of $\sqrt{n}$. Nevertheless, we omit this analysis. Finally, we can combine each Theorem 2, 3, 4 or 5 and Lemma 2 to construct an $\hat{\epsilon}$-KKT point of (1). Theorem 5 has a better complexity than Theorems 3 and 4, but requires stronger assumptions. Algorithm 2 is also different from the one in [3] both in terms of algorithmic form and the underlying problem to be solved, while achieving the same oracle complexity.

## 5 Numerical Experiments

We perform some experiments to illustrate Algorithm 1 and compare it with two existing and related algorithms. Further details and additional experiments can be found in Supp. Doc. D.

We consider the following regularized stochastic minimax problem studied, e.g., in [9, 33]:

$$\min_{w \in \mathbb{R}^p} \left\{ \max_{1 \leq j \leq m} \left\{ \frac{1}{n} \sum_{i=1}^n F_{i,j}(w) \right\} + \frac{\lambda}{2} \|w\|^2 \right\}, \tag{31}$$

where $F_{i,j} : \mathbb{R}^p \times \Omega \to \mathbb{R}_+$ can be viewed as the loss of the $j$-th model for data point $i \in [n]$. If we define $\phi_0(v) := \max_{1 \leq j \leq m}\{v_j\}$ and $f(w) := \frac{\lambda}{2}\|w\|^2$, then (31) can be reformulated into (3). Since $v_j \geq 0$, we have $\phi_0(v) := \max_{1 \leq j \leq m}\{v_j\} = \|v\|_\infty = \max_{\|u\|_1 \leq 1}\langle v, u \rangle$, which is nonsmooth. Thus we can smooth $\phi_0$ as $\phi_\gamma(v) := \max_{\|u\|_1 \leq 1}\{\langle v, u \rangle - (\gamma/2)\|u\|^2\}$ using $b(u) := \frac{1}{2}\|u\|^2$.

Here, we apply our problem (31) to solve a model selection problem in binary classification with nonnegative nonconvex losses, see, e.g., [41]. Each function $F_{i,j}$ belongs to 4 different nonconvex losses ($m = 4$): $F_{i,1}(w, \xi) := 1 - \tanh(b_i\langle a_i, w \rangle)$, $F_{i,2}(w, \xi) := \log(1 + \exp(-b_i\langle a_i, w \rangle)) - \log(1 + \exp(-b_i\langle a_i, w \rangle - 1))$, $F_{i,3}(w, \xi) := (1 - 1/(\exp(-b_i\langle a_i, w \rangle) + 1))^2$, and $F_{i,4}(w, \xi) := \log(1 + \exp(-b_i\langle a_i, w \rangle))$ (see [41] for more details), where $(a_i, b_i)$ represents data samples.

We implement 4 algorithms: our `SGM` with 2 options, `SGD` from [10], and `Prox-Linear` from [11]. We test these algorithms on two datasets from LIBSVM [6]. We set $\lambda := 10^{-4}$ and update the smooothing parameter $\gamma_t$ as $\gamma_t := \frac{1}{2(t+1)^{1/3}}$. The learning rate $\eta$ for all algorithms is finely tuned from $\{100, 50, 10, 5, 1, 0.5, 0.1, 0.05, 0.01, 0.001, 0.0001\}$, and the results are shown in Figure 1 for **w8a** and **rcv1** datasets using $k_b = 32$ blocks. The details of this experiment is given in Supp. Doc. D.

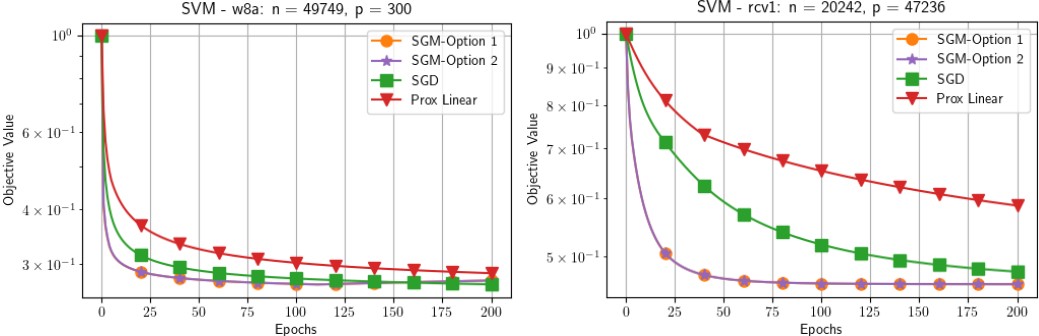

Figure 1: The performance of 4 algorithms for solving (31) on two datasets after 200 epochs.

As shown in Figure 1, the two variants of our `SGM` have a comparable performance with `SGD` and `Prox-Linear`, providing supportive evidence for using shuffling strategies in minimax algorithms.

## 6 Conclusions

This work explores a bilevel optimization approach to address two prevalent classes of nonconvex-concave minimax problems. These problems find numerous applications in practice, including robust learning and generative AIs. Motivated by the widespread use of shuffling strategies in implementing gradient-based methods within the machine learning community, we develop novel shuffling-based algorithms for solving these problems under standard assumptions. The first algorithm uses a non-standard shuffling strategy and achieves the state-of-the-art oracle complexity typically observed in nonconvex optimization. The second algorithm is also new, flexible, and offers a promising possibility for further exploration. Our results are expected to provide theoretical justification for incorporating shuffling strategies into minimax optimization algorithms, especially in nonconvex settings.

## Acknowledgments and Disclosure of Funding

This work was partly supported by the National Science Foundation (NSF): NSF-RTG grant No. NSF DMS-2134107 and the Office of Naval Research (ONR), grant No. N00014-23-1-2588.

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
