# OpenReview forum: "Shuffling Gradient-Based Methods for Nonconvex-Concave Minimax Optimization"
_NeurIPS.cc/2024/Conference — NeurIPS 2024 poster_

### Official Review · Reviewer_QEyv · 2024-06-20

**Soundness:** 3
**Presentation:** 3
**Contribution:** 3
**Rating:** 7
**Confidence:** 3

**Summary:**

This work propose shuffling gradient-based methods for nonconvex-linear and nonconvex-strongly convex minimax optimization and obtain complexity results.

**Strengths:**

The presentation is clear. Based on my knowledge on both shuffling and minimax optimization, I found that the algorithms and complexity results are reasonable. The experiments provide sufficient details for reproducibility.

**Weaknesses:**

The contribution is in general combinational, with slightly more techniques such as smoothing and shuffling-based function evaluations. Also, comparison with previous stochastic algorithms for minimax optimization and shuffling for nonconvex optimization is lacking, as shown in question 1 below.

**Questions:**

(1) You could compare your complexity results (Theorem 2) with previous stochastic gradient-based minimax optimization works (e.g. [a]) and shuffling for nonconvex optimization (e.g. [b]).

[a] Lin, T., Jin, C., \& Jordan, M. (2020, November). On gradient descent ascent for nonconvex-concave minimax problems. In International Conference on Machine Learning (pp. 6083-6093). PMLR.

[b] Nguyen, L. M., Tran-Dinh, Q., Phan, D. T., Nguyen, P. H., \& Van Dijk, M. (2021). A unified convergence analysis for shuffling-type gradient methods. Journal of Machine Learning Research, 22(207), 1-44.

(2) At the bottom of page 1, for problem (NL), it looks possible to remove matrix $K$ and use $F_i:\mathbb{R}^p\to \mathbb{R}^q$, since $\langle F_i(w),Ku\rangle=\langle K^{\top}F_i(w),u\rangle$ and we could replace $K^{\top}F_i(w)\in \mathbb{R}^q$ with $F_i(w)\in \mathbb{R}^q$.

(3) In Assumption 2, do you mean for any $v\in {\rm dom}(h)$, $||v||_2\le M_h$?

(4) In "Condition (6) can be relaxed to the form $\frac{1}{n} \sum_{i=1}^n||\nabla F_i(w)-\nabla F(w)||^2 \leq \sigma_J^2+\Theta_J||\nabla \Phi_0(w)||^2$", should $\Phi_0$ be $F$?

(5) After Assumption 5, ``If $f=0$, then $\mathcal{G} _ {\eta}(w)\equiv\nabla\Phi_{\gamma}(w)$''. Should it be $\nabla\Phi_0(w)$ instead of $\nabla\Phi_{\gamma}(w)$, since the definition of $\mathcal{G} _ {\eta}(w)$ in eq. (18) involves $\nabla\Phi_0(w)$ not $\nabla\Phi _ {\gamma}(w)$? Or you might consider defining $\mathcal{G} _ {\eta,\gamma}(w)$ with $\nabla\Phi_0(w)$ replaced by $\nabla\Phi _ {\gamma}(w)$, such that $\epsilon$-stationary point of (3) and that of (10) in Lemma 2 correspond to $\mathcal{G} _ {\eta,0}(w)$ and $\mathcal{G} _ {\eta,\gamma}(w)$ respectively.

(6) I found that Eq. (21) (Option 1) needs to save {$F_{\pi^{(t)}(j)(w_0^{(t)})}$}$_{j=1}^n$ to avoid repeated evaluation of these $n$ values. Yes?

(7) Line 255: "fix number" $\rightarrow$ "fixed number".

(8) At the end of Theorem 2, do you mean to change one of the $\nabla_w\mathcal{H}_i$ into $\nabla_u\mathcal{H}_i$? It seems that the complexity of semi-shuffling is bettter than full-shuffling. Then what's the advantage of full-shuffling?

**Limitations:**

The checklist mentions the limitation that this work only focuses on nonconvex-linear and nonconvex-strongly convex minimax optimization, and says "We do not yet know if our paper has an immediate broader impact. However, since our problems and our algorithms are sufficiently general, we hope they will create broader impacts." I agree with both of them.

---

> ### Author Rebuttal · Authors · 2024-08-01
>
> We thank you for your appreciation of our strengths and your positive evaluations. Please see our general responses along with the individual reply to you below.
>
> **The contribution ... below**
>
> We thank you for your comments. While we agree that some individual techniques in our paper are not new, we believe that our algorithms and problem setting is new and has not been addressed in prior works. Our shuffling approximation for the "hyper" gradient in the (NL) case is new. To our best knowledge, our methods for (NL) appear to be the first to develop shuffling-type methods for the class of nonconvex-linear minimax problems which has various applications in distributionally robust optimization and other learning scenarios (like risk-averse portfolio, model-agnostic meta learning).
> For the nonconvex-strongly concave setting (NC), we have two strategies, and our model is more general than that of (Cho & Yun, 2022) due to $f$ and $h$. Our assumptions and methods are also different from (Cho & Yun, 2022).
>
> **Comparisons of complexity**
>
> Thank you for your suggestion. We have a detailed comparison in the general response, which included the reference (Lin et. al., 2020):
>
> For (Nonconvex-SC/PL) setting ($f=h=0$), the complexity $\mathcal{O}(\sqrt{n} \epsilon^{-3})$ of our Algorithm 2 with a random reshuffling strategy and (Cho & Yun, 2022) is better than the complexity $\mathcal{O}(\epsilon^{-4})$ of non-shuffling iid scheme (Lin et. al., 2020, Theorem 4.5).
>
> About the paper (Nguyen et. el. 2021), although the setting is different, we will still compare with that reference in our paper.
>
> **Remove K**
>
> Yes, it is possible to remove $K$ as you suggested. Though, we note that $K$ plays an additional role as a linear transformation and matches the dimension between $u$ and $F(w)$ if they are not the same.
>
> **In Assumption 2 ...**
>
> Yes. That is correct.
>
> **In "Condition (6) ...?**
>
> It is $\nabla \Phi_0$ since our convergence guarantee is eventually on $\nabla \Phi_0$.
>
> **After Assumption 5 ...respectively.**
>
> Thank you for your suggestion. We will do as you suggested.
>
> **I found ... Yes?**
>
> That is correct. (21) saves the evaluation of $F_{\pi^{(t)}(j)}$, while (22) uses the most updated information of $F_{\pi^{(t)}(j)}$.
>
> **Line 255 ...**
>
> It will be fixed. Thank you.
>
> **At the end ... shuffling?**
>
> Yes, sorry for the typo, we meant that the complexity of $\nabla_w \mathcal H_i $ and $\nabla_u \mathcal H_i $ is $O ( \sqrt{n} \epsilon^{-3} )$. For the full shuffling, we believe that we can improve the complexity of $\nabla_u \mathcal H_i $ to match with the complexity of the semi-shuffling case, as discussed with reviewer jEDb. This would only be a minor change to the theoretical contributions of our paper.
>
> We hope our response answers all of your questions. If you have any additional comments and suggestions, please discuss with us and we are happy to clarify further.

---

> > ### Comment · Reviewer_QEyv · 2024-08-08
> > **Reviewer QEyv is satisfied with the authors' response and will increase rating to 7**
> >
> > Reviewer QEyv is satisfied with the authors' response and will increase rating to 7.

---

> > > ### Author Response · Authors · 2024-08-09
> > > **Thank you for your support!**
> > >
> > > Dear Reviewer QEyv,
> > >
> > > We are glad that you are satisfied with our response and increase the rating score. This is very encouraging!
> > >
> > > Thank you very much for your support!
> > >
> > > Best regards,
> > >
> > > Authors

---

### Official Review · Reviewer_jEDb · 2024-07-12

**Soundness:** 3
**Presentation:** 2
**Contribution:** 3
**Rating:** 5
**Confidence:** 3

**Summary:**

This paper focuses on nonconvex-concave, finite-sum (stochastic) minimax problems with possibly nonsmooth regularization. Aiming to find $\epsilon$-stationary points, the paper proposes shuffling-based proximal gradient descent-ascent algorithms and verifies gradient computation complexity upper bounds via a bilevel optimization perspective. The results consider various settings, such as (1) the nonconvex-linear (NL) case with either strongly convex or convex $h$, each requiring $\mathcal{O}(n \epsilon^{-3})$ and  $\mathcal{O}(n \epsilon^{-7/2})$ gradient evaluations, respectively, and (2) the nonconvex-strongly-concave (NS) case with smoothness assumptions, where semi-shuffling requires $\mathcal{O}(n \epsilon^{-3})$ while full-shuffling requires (naively) $\mathcal{O}(n \epsilon^{-4})$ gradient evaluations. The paper contains proofs of the convergence theorems and supporting numerical experiments.

**Strengths:**

* The theoretical results and the problem-specific proof techniques could be a solid contribution. To the best of my knowledge, this is one of the very few works that discuss shuffling-type algorithms for minimax problems apart from Das et al., 2022 [1] and Cho & Yun, 2023 [2]. I think the results under the relaxed setting, especially the linear, non-strongly-convex case of (NL), and for the bilevel-type shuffling algorithms are quite novel. The paper also includes experimental results that align with the theoretical results.

**Weaknesses:**

* Despite that the novelty of this work would be in the fact that the results apply for broader settings than previous literature rather than obtaining *faster* convergence under similar settings, the paper still should have included a fair, more detailed comparison with previous work and illustrated how the gradient evaluation complexity becomes different from settings with stronger assumptions. The current draft does not contain any meaningful quantitative comparison with previous work.
* In particular, I don’t think that the settings in [2] are different from the results for the (NS) setting. Apart from the fact that the objective function does not contain $f$ and $h$, the assumptions used in Theorem 1 of [1] (Assumptions 1-4) and the NS setting seem to be nearly identical, and the goal of finding $\epsilon$-stationary points also seem to be equivalent. (In fact, for the variable $u$ the PŁ assumption is weaker than strong concavity.) In terms of the convergence rates, Theorem 1 of [1] requires only $\mathcal{O} (n \epsilon^{-2} + n^{1/2} \epsilon^{-3})$ gradient evaluations, which is smaller than $\mathcal{O} (n \epsilon^{-3})$. The authors should have at least explained how introducing $f, h$ changes the difficulty of the problem, or given any other reasons why one could prefer the bilevel optimization framework over the simple SGDA algorithm as in [1].

**Questions:**

- Why is it necessary to use bilevel optimization, i.e., update $n$ steps of $w$ and then $n$ steps of $v$, instead of iteration-wise updates? Also, am I right that the results for the (NS) settings are weaker than that of [2] at least when $f, h = 0$? Is there a particular reason why the weaker convergence rates are inevitable if we consider $f, h$?
- Some technical parts of the algorithms are brought from previous work, such as the use of $\Phi_{\gamma}$ inspired by Yang et al., 2020 [3]. Some other parts look new, such as the placement of the proximal steps considering $f$ and $h$ and the variants of algorithms like choosing between (21), (22) in Algorithm 1 and (26), (27) in Algorithm 2. I am curious about the motivations of the new components of the algorithms, especially considering the different types of shuffling techniques I mentioned last—whether these have clear motivations or are more like artifacts of the proof techniques, if there were similar approaches in minimization problems, etc.

- Light question: Would it be able to relax the strongly concave case to weaker PŁ-type assumptions, or consider alternating type updates as in [2]? Moreover, is it possible to compare the presented results for minimax problems with nonconvex, nonsmooth *minimization* as in Cutkosky et al., 2023 [4]?
- Light question: Are there examples of when the nonconvex-linear (NL) setting could be of interest in practice?
- Minor: Why are there two bibliographies, and why are the contents different? It seems that the one at the end of the appendix has the correct numbers.

**Limitations:**

There seems to be no potential negative societal impact.

---

> ### Author Rebuttal · Authors · 2024-08-01
>
> We thank you for your appreciation of our contributions and your detailed reviews.  Please see our general responses along with the individual reply to you below.
>
> **Comparisons of complexity**
>
> Thank you. As you mentioned above, we found the three references [1] [2] and (Emmanouilidis et. al., 2024) that are most related to our paper. In our general response we compared them with our work in details (we do not repeat due to space limit).
>
> **In particular ... in [1]**
>
> We believe that you meant [2] (Cho & Yun, 2022). Thank you very much for this excellent point.
>
> We agree with you that the assumptions of Theorem 1 in [2] are similar to our assumptions for (NC) when $f=0$ and $h=0$ (the difference is Assumptions 3 + 4 of [2] and our strong concavity of $\mathcal{H}$). We believe that our analysis can be relaxed with Assumptions 3+4 in [2] by means of Eq. (21) in [2]). Let us explain in detail below.
>
> If we set S=1 in (27), then Alg. 2 performs only one shuffling loop to solve $\max_u$ subproblem. Alg. 2 is different from [2] due to epoch-wise, not update-wise as [2], but both have the same cost per epoch. When S is a constant and independent of $\eta$, we would have similar complexity as Theorem 1 of [2] if we use a random shuffling strategy. This was noted at the end of Theorem 4: a $\sqrt{n}$ improvement is achieved if random-reshuffling is used. Therefore, our complexity is $O (\sqrt{n} \epsilon^{-3} )$ as in [2].
>
> In our paper, the complexity of $\nabla_w \mathcal H_i $ could be $O ( \sqrt{n} \epsilon^{-3} )$, but the complexity of $\nabla_u \mathcal H_i $ is worse than than [2] by a factor of $1/\epsilon$. This is due to an overestimate of $S$ in Theorem 7, which depends on $1/\eta$, leading to a $1/\epsilon$ factor worse than [2]. This factor can be removed by using a similar Lyapunov func. $V_{\lambda}$ in [2] and modify our proof using Eq. (21) in [2] and Theorem 1 in [Nguyen et al, 2021] instead of our (56). We will update it in our revision.
>
> We believe that adding nonsmooth terms $f$ and $h$ could change the algorithm as one has to decide when to apply the prox operator, every single update or at the end of each shuffling loop. We are not sure how these terms could change the analysis of [2], including their assumptions (like our Ass 5). We also consider the strong-concavity either of $\mathcal{H}$ or of $-h$ separately (resp., our Thm 3 or 4).
>
> We think that the bilevel opt. approach has more flexibility to tackle the $\max$ and the $\min$ independently, while the SGDA tackles them simultaneously. In general [not only specific to [2], since [2] looks like a mixed approach], the bilevel optimization approach also allows one to explore techniques from standard optimization, while Nash’s game approach (simultaneously solve the min and the max) such as SGDA often requires theory from montone/nonmonotone operators.
>
> Apart from the above points, our work also considers the nonconvex-linear case (NL), overlapped with (NC) in Algorithm 1 (a new shuffling rule like (22)), and a semi-shuffling strategy, which could also be useful in practice.
>
> **Why is ... $f$ and $h$?**
>
> For the choice of bilevel optimization approach, we have explained above. It is not necessary, but just a different approach. Note that Algorithm 2 only proposed two options (26) and (27), but other options (like accelerated or momentum gradient methods) may be possible to use to solve the strongly concave subproblem $\max_u$.
>
> For $f, h=0$, our complexity of $\nabla_w\mathcal{H}_i$ is the same as in Th. 1 of [2], but the complexity of $\nabla_u\mathcal{H}_i$ is currently weaker than Th. 1 of [2]. As we explained above, we can improve the latter one to $O(\sqrt{n}\epsilon^{-3})$.
>
> **Some ... etc.**
>
> Yes, we have a clear motivation. For (NL), it is overlapped with (NC), but one is not included in the other. This setting is important as it can be reformulated equivalently to the compositional minimization (CO), and has many applications, including distributionally robust optimization (see Q4 below). The choice of (21) saves evaluations of $F_i$, while (22) uses the most updated information of $F_i$.
>
> For (NC), we are flexible to use different methods to solve the strongly concave subproblem $\max_u$. We gave two options: (26) and (27), but other choices can be used.
>
> When the maximization is "simple", it may be better to use a semi-shuffling strategy (26), where we can solve the max problem by deterministic methods. Note that one can replace (26) with other deterministic methods, but we need to carry out different analyses.
>
> Furthermore, we believe that the strong convexity either in the smooth term $\mathcal{H}$ or in the nonsmooth term $h$ also makes a difference. So, we cannot incorporate $f$ and/or $h$ in the smooth term $\mathcal{H}$ if they are nonsmooth.
>
> For the motivation of the algorithm, we aim to design a natural application of shuffling data schemes in solving the stated minimax problems. The analyses of our algorithms follow after we have tried several different proof techniques.
>
> **Light question: Would ... [4]?**
>
> As mentioned above, our analysis still works with Assumptions 3+4 in [2] by means of Eq. (21) in [2], at least when $f=0$ and $h=0$. This expression can be written as $L_u \Vert \tilde{u}_t - u^{*} ( \tilde{w} _{t-1} ) \Vert^2 \leq 2\kappa_2 [ \Phi_0 ( \tilde{w} _{t-1} ) - \mathcal{H} (\tilde{w} _{t-1}, \tilde u_t ) ] $ in our context. We just need to adapt our proof lightly to accommodate it.
>
> **Light question: Are ... practice?**
>
> Yes, there are so many applications, including distributionally robust optimization with finite discrete probability distribution, risk-averse portfolio, model-agnostic meta learning [Finn et al (2017)], etc.
>
> **Minor: Why ... numbers?**
>
> We are sorry about this technical issue. We will fix it.
>
> We hope our response answers all of your concerns and questions. If you have any additional comments, please discuss with us and we are happy to clarify further.

---

> > ### Author Response · Authors · 2024-08-09
> > **Follow up on the rebuttal**
> >
> > Dear Reviewer jEDb,
> >
> > We hope our responses answer all your questions!
> >
> > In case you need any remaining clarifications, we would be more than happy to reply. Please let us know your thoughts as soon as you can (within this discussion period). If your questions are all properly addressed, we really hope that you consider increasing your score to support our work.
> >
> > Regards,
> >
> > Authors

---

> > > ### Comment · Reviewer_jEDb · 2024-08-11
> > >
> > > Thank you for your detailed and precise answers.

---

> > > > ### Author Response · Authors · 2024-08-11
> > > > **Thank you!**
> > > >
> > > > Dear Reviewer jEDb,
> > > >
> > > > We thank you very much for your time and your appreciation. Your support is meaningful for us to explore this direction.
> > > >
> > > > Thank you again!
> > > >
> > > > Best regards,
> > > >
> > > > Authors

---

### Official Review · Reviewer_czC8 · 2024-07-12

**Soundness:** 3
**Presentation:** 3
**Contribution:** 2
**Rating:** 5
**Confidence:** 2

**Summary:**

This paper proposes shuffling gradient-based methods for addressing two classes of minimax optimization problems: nonconvex-linear and nonconvex-strongly concave settings. The first algorithm focuses on the nonconvex-linear minimax setting and the second algorithm, consisting of semi-shuffling and full-shuffling schemes, focuses on the nonconvex-strongly concave minimax setting. The authors establish oracle complexity bounds for these algorithms under some standard assumptions. Numerical experiments are conducted.

**Strengths:**

The authors establish oracle complexity bounds for their algorithms under standard assumptions. These bounds provide theoretical guarantees on the performance of the proposed methods.

**Weaknesses:**

The proposed Algorithm 1 requires solving a maximization problem at each iteration, which can be computationally expensive.

The numerical experiment section is quite weak. First, only Algorithm 1 is tested, with no results related to Algorithm 2. Second, only one minimax model is tested. More complex and popular minimax models should be included, and state-of-the-art competitors should be used to evaluate the performance of the proposed methods.

Typo in Line 187 and Line 459: $M_f \rightarrow M_f^2$

**Questions:**

What are the advantages of the proposed methods in this paper, given the introduction of the semi-shuffling and full-shuffling schemes? Do they have a better convergence rate or sample complexity compared to existing methods?

**Limitations:**

yes

---

> ### Author Rebuttal · Authors · 2024-07-31
>
> We thank you for your appreciation of our strengths and soundness. Please see our responses to your concerns and questions below:
>
> **The proposed ... expansive**
>
> We can choose a simple $b$ in (9) so that $u^*_{\gamma} ( \cdot )$ in Step 9 of Algorithm 1 has a **closed-form solution**. For instance, if $b(u) = \frac{1}{2}\Vert u - u^0\Vert^2$ for a fixed $u^0$, then computing $u^*_{\gamma}(\cdot)$ reduces to computing the proximal operator of $h$ plus a linear term, i.e. $u^*_{\gamma}( w ) := \mathrm{prox}_{h/\gamma}(u^0 - K^{\top}F(w)/\gamma)$. For many $h$ (e.g., $\ell_1$-norm, the indicator of simple sets), this proximal operator has a closed form.
>
>
> **The numerical ... methods**
>
> We appreciate your point. We can add more numerical examples in the Supp. Doc. as you suggested.
>
> Apart from this point, we believe that our algorithms and theoretical analysis deserve to be reconsidered since we have 4 theorems corresponding to 5 settings (2 settings for (NL) and 3 ones for (NC)). To our best knowledge, our methods for (NL) appear to be the first to develop shuffling-type methods for the class of nonconvex-linear minimax problems which has various applications in distributionally robust optimization and other learning scenarios (like risk-averse portfolio, model-agnostic meta learning). The shuffling strategy we use in Algorithm 1 is also new.
>
> For the nonconvex-strongly concave setting (NC), we have two strategies, and our model is more general than that of [9] due to $f$ and $h$. Our assumptions and methods are also different from [9].
>
> **Typo**
>
> Thank you. We will fix it.
>
> **What are the advantages**
>
> In our general response, we showed the complexity comparisons for our paper, which highlight the advantages of our shuffling method and settings compared to other prior work. A comparison between shuffling and SGD was done in (Cho & Yun, 2022), showing improvement of complexity under concrete scenarios.
>
> As discussed in the first paragraph of the introduction, the motivation to study shuffling methods is that they are widely implemented in well-established packages like TensorFlow and Pytorch for optimization and deep learning. Empirically, it was observed, see e.g. [(Bottou, 2009, 2012; Hiroyuki, 2018)] that shuffling schemes decrease the loss faster than SGD and work well in practice, and in many cases. Such aspects motivate us to propose shuffling methods for minimax problems. We are not aware of a rigorously empirical study for minimax problems since such methods have not widely been developed yet. However, many researchers have used SGD to train GANs through standard libraries like TensorFlow or Pytorch, we believe that they could use shuffling SGD implemented in these packates. Our paper studies theoretical convergence guarantees for such strategies in minimax algorithms.
>
> We hope our response answers all of your concerns and questions. If you have any additional comments and suggestions, please discuss with us and we are happy to clarify further.

---

> > ### Author Response · Authors · 2024-08-09
> > **Follow up on the rebuttal**
> >
> > Dear Reviewer czC8,
> >
> > We hope our responses answer all your questions!
> >
> > In case you need any remaining clarifications, we would be more than happy to reply. Please let us know your thoughts as soon as you can (within this discussion period). If your questions are all properly addressed, we really hope that you consider increasing your score to support our work.
> >
> > Regards,
> >
> > Authors

---

> > > ### Author Response · Authors · 2024-08-11
> > > **Follow up**
> > >
> > > Dear Reviewer czC8,
> > >
> > > Since the Author-Reviewer discussion phase will end in two days, we would like to follow up and discuss with you. Please do not hesitate to contact us if there are additional answers or explanations that we can make to clarify our paper within this discussion period. We appreciate your timely response, as it would provide us with an opportunity to address any remaining questions.
> > >
> > > If your concerns are all properly addressed, we really hope that the reviewer positively re-evaluates our work. We appreciate your inputs and we thank you for your time spent reviewing this paper.
> > >
> > > Best regards,
> > >
> > > Authors

---

> > > > ### Comment · Reviewer_czC8 · 2024-08-13
> > > >
> > > > Thank you for the detailed response to my questions. I will raise my score accordingly.

---

> > > > > ### Author Response · Authors · 2024-08-13
> > > > > **Thank you very much for your support!**
> > > > >
> > > > > Dear Reviewer czC8,
> > > > >
> > > > > Thank you very much for your feedback and support!
> > > > >
> > > > > Best regards,
> > > > >
> > > > > Authors

---

### Official Review · Reviewer_4w6E · 2024-07-13

**Soundness:** 3
**Presentation:** 3
**Contribution:** 3
**Rating:** 6
**Confidence:** 3

**Summary:**

The paper presents new shuffling gradient-based methods for solving two classes of nonconvex-concave minimax optimization problems: nonconvex-linear and nonconvex-strongly concave settings.

The first algorithm is designed for the nonconvex-linear setting and achieves state-of-the-art oracle complexity, employing a new shuffling estimator for the hyper-gradient.

The second method introduces semi-shuffling and full-shuffling schemes for the nonconvex-strongly concave setting, establishing their oracle complexity bounds for the first time.

Numerical examples are provided to demonstrate the performance of the proposed algorithms.

**Strengths:**

1. The paper is well-written, offering a clear exposition of the current state of the art, along with explicit assumptions and conditions for the proposed methods.

2. The authors introduce novel smoothing techniques similar to Nesterov's smoothing approach for lower-level maximization problems, and establish oracle complexity bounds for both nonconvex-linear and nonconvex-strongly concave settings. This introduces a novel perspective on addressing nonconvex optimization problems.

3. The paper presents two distinct algorithms, each carefully tailored to different problem settings, showcasing a profound comprehension of diverse problem landscapes.

**Weaknesses:**

1. **Complexity Comparison with Existing Works**. The authors are encouraged to present a comparative table that delineates the oracle complexity of their proposed methods alongside that of existing works, including standard nonshuffling methods. This side-by-side comparison should aim to explicitly demonstrate the theoretical advantages of their methods.

2. **Shuffling Methods vs. Nonshuffling Methods**. The paper should discuss the distinctions between shuffling methods and traditional nonshuffling approaches. Specically, it is better to clarify whether shuffling methods consistently yield superior performance over nonshuffling methods theoretically.

3. **Numerical Experiments Expansion**. While the experiments have been conducted on synthetic datasets and small-scale datasets, which is customary for theoretical papers, the scope of empirical validation could be enhanced. It would be much better if the authors could extend their experiments to include large-scale datasets.

**Questions:**

See above.

**Limitations:**

Yes

---

> ### Author Rebuttal · Authors · 2024-07-31
>
> We thank you for your appreciation of our strengths and your positive evaluations. We addressed your comments in the general response. We repeat it below for your convenience:
>
> **1. Comparison of complexity with other shuffling methods**
>
> Thank you for your suggestions. Among the stochastic shuffling gradient methods for the minimax problem, we have found three references (Das et. al., 2022; Cho & Yun, 2022; Emmanouilidis et. al., 2024) that are most related to our paper.
>
> However, the two references (Das et. al., 2022; Emmanouilidis et. al., 2024) are different from us that they consider two-sided PL condition for $H(w,u)$, or strongly monotone variational inequality. These settings are stronger than our general nonconvexity of $H_i$ in the first variable $w$. Thus our results cannot obtain the strongly convex (or PL) rate of $\mathcal{O}(\frac{1}{nK^2})$ as in these references.
>
> In comparison with the reference (Cho & Yun, 2022), our setting is broader than the problem in (Cho & Yun, 2022) that we consider functions $f$ and $h$. When $f = h = 0$, the semi-shuffling variant of Algorithm 2 with a random reshuffling strategy obtains the complexity of $\mathcal{O}(\sqrt{n} \epsilon^{-3})$ (noted in line 300, page 9), which is comparable with the complexity of Theorem 1 (Nonconvex-PL) in (Cho & Yun, 2022).
>
> Note that our Algorithm 2 is different from (Cho & Yun, 2022) even when $f=h=0$ since we do alternating epoch-wise update, while (Cho & Yun, 2022) uses component-wise update (alternating or simultaneous). When $f$ and/or $h$ are present, it remains unclear how to incorporate the prox operators in (Cho & Yun, 2022) as we have done.
>
> In addition, our analysis can be relaxed to the settings of the PL Assumptions 3+4 in (Cho & Yun, 2022) and obtain the same complexity as the strongly convex case.
>
> **2. Comparison of complexity with other non-shuffling methods**
>
> A comparison between shuffling and other methods, including non-shuffling schemes was done in (Cho & Yun, 2022). For (Nonconvex-SC/PL) setting, the complexity $\mathcal{O}(\sqrt{n} \epsilon^{-3})$ of our Algorithm 2 with a random reshuffling strategy and (Cho & Yun, 2022) is better than the complexity $\mathcal{O}(\epsilon^{-4})$ of non-shuffling iid scheme (Lin et. al., 2020, Theorem 4.5).
>
> Not only for minimax optimization, but it has also been observed that shuffling gradient methods often have faster convergence than non-shuffling versions for stochastic optimization (Nguyen et. al. 2021). Moreover, in practice, these methods have shown improved performance over iid algorithms (Bottou, 2009, 2012; Hiroyuki, 2018).
> This is the motivation for us to work on the broad setting of our paper.
>
> All the discussions above will be added to our latest revision, and we will also add complexity comparison table.
>
> **3. Experiments**
>
> We thank you for your suggestions. We will add more examples in the revision. We tested our method on a relatively large dataset *url* from LibSVM with n=2,396,130 and p=3,231,951 for our binary classification. The results are in the appendix.
>
> However, we would like to emphasize that the main contributions of our paper are theoretical. We have 4 theorems corresponding to 5 settings (2 settings for (NL) and 3 ones for (NC)). To our best knowledge, our methods for (NL) appear to be the first to develop shuffling-type methods for the class of nonconvex-linear minimax problems which has various applications in distributionally robust optimization and other learning scenarios (like risk-averse portfolio, model-agnostic meta learning). The shuffling strategy we use in Algorithm 1 is also new.
>
> For the nonconvex-strongly concave setting (NC), we have two strategies, and our model is more general than that of (Cho & Yun, 2022) due to $f$ and $h$. Our assumptions and methods are also different from (Cho & Yun, 2022).
>
> Overall, our paper is significant different from prior works except for (Cho & Yun, 2022) which we discussed above. We have other main contributions for the (NL) case, and our semi-shuffling scheme, apart from handling the nonsmooth terms $f$ and/or $h$.
>
>
> We hope our response answers all of your questions. If you have any additional comments and suggestions, please discuss with us and we are happy to clarify further.
>
> Again, thank you for your efforts in reviewing this paper!

---

### Author Rebuttal · Authors · 2024-08-02

Dear Reviewers,

Thank you all so much for reviewing our submission. We valued your comments and appreciation of our strengths and contributions.

We have addressed each comment individually for each reviewer. In this general response, we highlight some common responses to all the reviewers.

**1. Comparison of complexity with other shuffling methods**

Thank you for your suggestions. Among the stochastic shuffling gradient methods for the minimax problem, we have found three references (Das et. al., 2022; Cho & Yun, 2022; Emmanouilidis et. al., 2024) that are most related to our paper.

However, the two references (Das et. al., 2022; Emmanouilidis et. al., 2024) are different from us that they consider two-sided PL condition for $H(w,u)$, or strongly monotone variational inequality. These settings are stronger than our general nonconvexity of $H_i$ in the first variable $w$. Thus our results cannot obtain the strongly convex (or PL) rate of $\mathcal{O}(\frac{1}{nK^2})$ as in these references.

In comparison with the reference (Cho & Yun, 2022), our setting is broader than the problem in (Cho & Yun, 2022) that we consider functions $f$ and $h$. When $f = h = 0$, the semi-shuffling variant of Algorithm 2 with a random reshuffling strategy obtains the complexity of $\mathcal{O}(\sqrt{n} \epsilon^{-3})$ (noted in line 300, page 9), which is comparable with the complexity of Theorem 1 (Nonconvex-PL) in (Cho & Yun, 2022).

Note that our Algorithm 2 is different from (Cho & Yun, 2022) even when $f=h=0$ since we do alternating epoch-wise update, while (Cho & Yun, 2022) uses component-wise update (alternating or simultaneous). When $f$ and/or $h$ are present, it remains unclear how to incorporate the prox operators in (Cho & Yun, 2022) as we have done.

In addition, our analysis can be relaxed to the settings of the PL Assumptions 3+4 in (Cho & Yun, 2022) and obtain the same complexity as the strongly convex case.

**2. Comparison of complexity with other non-shuffling methods**

A comparison between shuffling and other methods, including non-shuffling schemes was done in (Cho & Yun, 2022). For (Nonconvex-SC/PL) setting, the complexity $\mathcal{O}(\sqrt{n} \epsilon^{-3})$ of our Algorithm 2 with a random reshuffling strategy and (Cho & Yun, 2022) is better than the complexity $\mathcal{O}(\epsilon^{-4})$ of non-shuffling iid scheme (Lin et. al., 2020, Theorem 4.5).

Not only for minimax optimization, but it has also been observed that shuffling gradient methods often have faster convergence than non-shuffling versions for stochastic optimization (Nguyen et. al. 2021). Moreover, in practice, these methods have shown improved performance over iid algorithms (Bottou, 2009, 2012; Hiroyuki, 2018).
This is the motivation for us to work on the broad setting of our paper.

All the discussions above will be added to our latest revision, and we will also add complexity comparison table.

**3. Motivation (Reviewer #jEDb).**
As we will explaine below, we have clear motivation when combining different components. Some are due to different assumptions (e.g., strongly concave vs merely concave), and some are optional (e.g. semi-shuffling vs full-shuffling). In addition, our shuffling strategy is quite general, which can cover both deterministic (e.g., incremental methods) and randomized variants.

**4. Experiments**

We thank you for your suggestions. We will add more examples in the revision. We tested our method on a relatively large dataset *url* from LibSVM with n=2,396,130 and p=3,231,951 for our binary classification. The results are in the appendix.

However, we would like to emphasize that the main contributions of our paper are theoretical. We have 4 theorems corresponding to 5 settings (2 settings for (NL) and 3 ones for (NC)). To our best knowledge, our methods for (NL) appear to be the first to develop shuffling-type methods for the class of nonconvex-linear minimax problems which has various applications in distributionally robust optimization and other learning scenarios (like risk-averse portfolio, model-agnostic meta learning). The shuffling strategy we use in Algorithm 1 is also new.

For the nonconvex-strongly concave setting (NC), we have two strategies, and our model is more general than that of (Cho & Yun, 2022) due to $f$ and $h$. Our assumptions and methods are also different from (Cho & Yun, 2022).

Overall, our paper is significant different from prior works except for (Cho & Yun, 2022) which we discussed above. We have other main contributions for the (NL) case, and our semi-shuffling scheme, apart from handling the nonsmooth terms $f$ and/or $h$.

**Finally,**

We hope our response answers all of your concerns and questions. If you have any additional comments and suggestions, please discuss with us and we are happy to clarify further.

Again, thank you for your efforts in reviewing this paper!

**Added References:**

Tianyi Lin, Chi Jin, and Michael Jordan. On gradient descent ascent for nonconvex-concave minimax problems.  In International Conference on Machine Learning, pp. 6083–6093. PMLR, 2020

L. Bottou. Curiously Fast Convergence of Some Stochastic Gradient Descent Algorithms. In Proceedings of the Symposium on Learning and Data Science, Paris, volume 8, pages 2624–2633, 2009

L. Bottou. Stochastic Gradient Descent Tricks. In Neural networks: Tricks of the trade, pages 421–436. Springer, 2012.

K. Hiroyuki. SGDLibrary: A MATLAB Library for Stochastic Optimization Algorithms. Journal of Machine Learning Research, 18(215):1–5, 2018.

---

### Author Response · Authors · 2024-08-14
**Thank you!**

Dear Area Chair and Reviewers,

We would like to thank the AC and the reviewers for your time and effort in handling and reviewing our paper.

Best regards,

Authors

---

### Decision · Program_Chairs · 2024-09-25

**Decision:**

Accept (poster)

**Comment:**

Stochastic methods for minimax optimization using traditional IID sampling strategies are well-studied. There is less work on shuffling methods for minimax optimization which may perform better in practice. The authors design shuffling based algorithms for nonconvex-linear minimax and nonconvex-strongly concave minimax problems, and provide worst-case analysis of their proposed algorithms for finding stationary points. In the former setting their algorithm matches the best-known oracle complexities (achieved with IID sampling strategies). The authors tested one of their algorithms numerically. While it performed well in the author’s experiments, the test problems could have been more compelling and the competing algorithms stronger.

Comments for the authors:

While you state the minimax setting is well-studied, it would be helpful if you provided a paragraph in the related work pointing to papers that study the NL and NC setting using IID sampling strategies. In particular, it would have been nice if there were clear comparisons between the techniques. A table as reviewer 4w6E suggested would be helpful as well.

Please be careful about how you describe IID sampling strategies, sometimes you describe it as SGD which is confusing. Typically people do not use SGD in this way. For example, often people say “SGD with Random Shuffling”, etc. In particular, one page the phrase “that the new methods achieve comparable performance to SGD” should be rephrased as our shuffling based algorithm achieves comparable performance to methods that use IID sampling strategies. This is an issue throughout the paper including in the labels on the numerical results.

Phi_0 is introduced on page 2 with no description but not defined until page 3